# Effect of Basalt Fiber on Uniaxial Compression-Related Constitutive Relation and Compressive Toughness of Recycled Aggregate Concrete

**DOI:** 10.3390/ma16051849

**Published:** 2023-02-23

**Authors:** Yaodong Guo, Yuanzhen Liu, Wenjing Wang, Kaidi Wang, Yu Zhang, Jingguang Hou

**Affiliations:** 1Department of Civil Engineering, Taiyuan University of Technology, Taiyuan 030000, China; 2School of Environmental and Safety Engineering, North University of China, Taiyuan 030000, China; 3Faculty of Engineering, China University of Geosciences, Wuhan 430000, China

**Keywords:** basalt fiber-reinforced recycled aggregate concrete (BFRAC), fiber strengthening factor, complete stress–strain curve model, toughness

## Abstract

The deformation performance of recycled aggregate concrete can be effectively improved when basalt fiber is reasonably added. In this paper, the effects of the basalt fiber volume fraction and the length–diameter ratio on the uniaxial compression-related failure characteristics, feature points of the complete stress–strain curve and the compressive toughness of recycled concrete under different replacement rates of recycled coarse aggregate were studied. The results showed that with the increase in the fiber volume fraction, the peak stress and peak strain of basalt fiber-reinforced recycled aggregate concrete first increased and then decreased. With the increase in the fiber length–diameter ratio, the peak stress and strain of the basalt fiber-reinforced recycled aggregate concrete first increased and then decreased, whereas the effect of the length–diameter ratio on peak stress and strain of the basalt fiber-reinforced recycled aggregate concrete was clearly smaller than that of the fiber volume fraction. Based on the test results, an optimized stress–strain curve model of concrete under uniaxial compression was proposed for the basalt fiber-reinforced recycled aggregate concrete. Furthermore, it was found that the fracture energy is more suitable for evaluating the compressive toughness of the basalt fiber-reinforced recycled aggregate concrete than the tensile–compression ratio.

## 1. Introduction

Recycled aggregate concrete (RAC) refers to a type of green and low-carbon concrete [1] which takes on critical significance for achieving the goals of recycling and sustainable development of waste concrete. However, the recycled coarse aggregate processed from waste concrete has defects (e.g., micro-cracks and holes) such that the mechanical properties and durability of RAC are inferior to those of ordinary concrete [2,3,4]. Existing research has confirmed that the brittle failure characteristic of RAC is more significant than that of ordinary concrete [5,6], and it tends to be more obvious with the increase in the replacement rate of the recycled coarse aggregate [6].

Concrete toughening technology includes base toughening and fiber toughening [7]. Base toughening mainly optimizes the hydration products, while fiber toughening is used to obtain the composite material through physical bridging technology. When compared with base toughening, fiber toughening technology is simple and feasible. Currently, a variety of fibers have been applied in concrete and achieved a positive effect in toughening and cracking resistance. Among these, steel fiber is the most popular fiber material and the most used in structures [8]. However, steel fiber increases the weight of the structure and reduces the workability of the mixture; its corrosion resistance is also poor. Basalt fiber (BF), a new type of inorganic and environmentally friendly fiber, has become one of the current research hotspots due to its light weight, strong corrosion resistance [9], high tensile strength and good compatibility with concrete [10]. By adding randomly distributed BF to RAC to prepare basalt fiber-reinforced recycled aggregate concrete (BFRAC), the effect of BF on RAC toughness and resistance cracking can be fully obtained, and the adverse effect of recycled coarse aggregate (RCA) on concrete performance can be compensated.

Currently, the research on BF for the improvement of concrete toughness focuses on the influence of BF volume content, length and other influencing factors on concrete compressive toughness, tensile toughness and flexural toughness. Wang Yi [11] took the area bounded by the uniaxial compression load–displacement curve as the toughness index; they studied the influence of different fiber lengths and different fiber volume fractions on the compressive toughness of concrete. The results show that the average increase rate of toughness of BFRC is 63% when compared with NC. Guo Yaodong [12] studied the influence of the basalt fiber volume fraction and the length–diameter ratio on the axial tensile properties of concrete; the study found that the maximum increase rate of the tensile toughness of BFRC when compared with NC was 43.0% according to the index of uniaxial tensile failure fracture energy.

Wang Qi [13] took the area bounded by the load–deflection curve when the peak load deflection was reached as the index to study the influence of basalt fiber content and length on concrete flexural toughness. The results show that the maximum increase rate of BFRC flexural toughness is 53.18% when compared with NC. DIAS et al. [14] studied the influence of BF content on the flexural properties of NC and polymer concrete through a three-point flexural test. It was found that BF can significantly improve the toughness of concrete. All the above studies show that BF has a positive effect on the improvement of concrete toughness. Based on the complete stress–strain curve test of BF rubber RAC under uniaxial compression, Chen Meng [15] adopted the curve simulation using the Guozhenhai model, established the complete stress–strain curve equation applicable to BF rubber RAC and analyzed its deformation performance under compression. However, the research on the stress–strain constitutive relationship of BFRAC under uniaxial compression is rarely reported. At the moment, research on BFRAC mainly focuses on the influence of a single factor (BF content) on the basic mechanical properties of RAC. The literature review by Elshafle and Whittleston [16] showed that BF lengths ranging from 12 mm to 24 mm and volume content ranging from 0.1% to 0.5% can improve the basic mechanical properties of concrete. Mehran Khan [17] studied the effect of BF length on steel fiber calcium carbonate crystal-reinforced fly ash concrete. The results show that the effect of 12 mm of BF on the compressive properties of fly ash concrete is generally better than that of longer fiber. These results reveal that the length–diameter ratio of BF significantly affects the mechanical properties of concrete.

In summary, this paper explored the effects of the BF volume fraction (0, 0.05%, 0.10%, 0.15%, 0.20%, 0.25%, 0.30% and 0.35%) and length–diameter ratio (1000, 1200, 1400 and 1600) on the deformation performance of RAC under compression through the uniaxial compression test. Based on the complete stress–strain curve, the effect of BF characteristic parameters on the compressive toughness of RAC was analyzed; this provides a reference for the current engineering design of BFRAC.

## 2. Test Scheme

### 2.1. Materials

The cementitious materials used in this test were ordinary Portland cement (P·O 42.5) and grade I silica fume (SF). The main chemical composition and physical properties of cement and SF are shown in Table 1. Crushed stone was used as natural coarse aggregate (NCA), with a particle size of 5~20 mm. The recycled coarse aggregate was broken from the concrete with an original design strength grade of C40, and the particle size was 5~20 mm. Its grading curve is shown in Figure 1. Fine aggregate (S) was made from natural river sand with a particle size of 0.075~4.75 mm. The physical properties of the aggregate are shown in Table 2. The test water was tap water. A high-efficiency polyhydroxy acid water-reducing agent was used as the admixture; it had a water reducing rate of 23~25%. The diameter of BF monofilament was 15 μm. The physical indexes are shown in Table 3.

### 2.2. Mix Proportion

The design strength grade of the base concrete was C40, and the specific mix proportion is shown in Table 4. The BF volume fraction (*V*_f_) is the percentage of BF volume to the volume of BFRAC. The length–diameter ratio (*l*/*d*) is the ratio of the fiber length (*l*) to the fiber diameter (*d*). The replacement rate of recycled coarse aggregate (*r*) is the percentage of the volume of recycled coarse aggregate in the volume of coarse aggregate. The specimen numbers were grouped according to Table 5.

### 2.3. Specimen Making

The size, number and curing method of the test specimen were in accordance with the “Standard Test Method for Physical and Mechanical Properties of Concrete” (GB/T 50081-2019). The size of the compressive strength test block was 100 mm × 100 mm × 100 mm, and the size of prismatic specimen was 150 mm × 150 mm × 300 mm for the uniaxial compression test; three samples were made for each group. In order to improve the dispersion performance of the BF in the concrete, the BF was pretreated at a high temperature (300 °C), and a variety of mixing methods were compared and analyzed. The mixing method shown in Figure 2 was used. After 24 h of mold removal, all specimens were placed into the standard curing tank for curing. The temperature was controlled at 20 °C, and the relative humidity was greater than 95%; the curing age was 28 days.

### 2.4. Test Method

The cube compressive strength test was performed in accordance with the specification “Standard Test Method for Physical and Mechanical Properties of Concrete” (GB/T 50081-2019). In order to obtain a more ideal complete stress–strain curve under axial compression, a rigid auxiliary element was added to the press in this test. In the elastic stage, the force was used to control the loading. The loading speed was 0.8 KN/s; this was 80% of the axial compressive strength of concrete. In the convex ordinate of the ascending curve, displacement control was used; the loading speed was 0.002 mm/s. When the residual strength of the descending curve was 60% of the axial compressive strength, the displacement control loading rate was adjusted to 0.004 mm/s. When the load was 30% of the peak load, the test stopped. The test device is shown in Figure 3.

## 3. Test Results and Analysis

### 3.1. Basic Mechanical Properties

#### 3.1.1. Compressive Strength

(1) Effect of fiber volume fraction

Figure 4 shows the relationship between the compressive strength of the BFRAC cube and the BF volume fraction. As depicted in Figure 4, when the RCA replacement rate was 0, 50% and 100%, the compressive strength of the BFRAC cube increased first and then decreased with the increase in BF volume fraction. This trend did not change with the length–diameter ratio. By comparing and analyzing the relationship between the compressive strength of BFRAC and the BF volume fraction at different replacement rates, it can be observed that the optimal volume fraction and the optimal length–diameter ratio of BFRAC are slightly different. The optimal volume fraction was 0.20~0.25%, and the optimal length–diameter ratio was 1200~1400. It is worth noting that the compressive strength of BF3516R0 was 3.06% lower than that of ordinary concrete, i.e., the fiber volume fraction was too large; this had an adverse effect on the compressive strength of NC. However, the RAC did not show a negative effect in the experimental dosage range.

By analyzing the increase rate of the BF on the compressive strength of the NC and the RAC, it was found that the maximum increase rate of BF on the compressive strength of the RAC was greater than that of the NC. When the length–diameter ratio was 1200 and the volume fraction was 0.20%, the compressive strength of the RAC with the replacement rates of 50% and 100% increased by 18.31% and 24.0%, respectively, when compared with that of non-doped RAC. However, Liu’s results showed that the best compressive strength improvement was achieved when the fiber volume fraction was 0.2%. However, the increase rate was only 8% which may be caused by uneven fiber dispersion [18]. This is mainly because the “reservoir effect” of RCA and the “bridging effect” of BF have synergistic effects on RAC. The schematic diagram of the “reservoir effect” of RCA is shown in Figure 5. As depicted in this figure, in the BFRAC, the fiber and the base jointly bore the load, and the stress was conducted through the Interfacial Transition Zone (ITZ) between the fiber and the base. At the same time, due to the bridging effect of fiber, the propagation of primary cracks in concrete was constrained and limited. With the hydration of cementitious materials, the interfacial transition zone near the fiber surface had higher porosity and more defects were formed. Due to the presence of old mortar attached to the surface of the RCA and internal micro-cracks, the water absorption rate of the RCA was higher. In the mixing process, the RCA absorbed more free water inside the concrete and played the role of a “reservoir” inside the concrete. In the later curing period of concrete, the free water was gradually released to form an “inner curing” for the RAC, thus promoting the hydration of unhydrated cement particles and further improving the hydration degree. It also effectively reduced the porosity of the interface transition zone between RCA and BF, which improved the bond performance between BF and interface transition zone, as well as the strength of the RAC.

(2) Effect of fiber length–diameter ratio

Figure 6 shows the relationship between the compressive strength of the BFRAC cube and the BF length–diameter ratio. As depicted in Figure 6, the effect of the BF length–diameter ratio on the compressive strength of RAC was consistent with that of ordinary concrete. On the whole, with the increase in the BF volume fraction, it first increased and then decreased.

The difference was that for the BFRAC of recycled coarse aggregate with the replacement rate of 100%, when the fiber volume fraction of BFRAC was less than 0.15%, the compressive strength of the BFRAC decreased with the increase in the BF length–diameter ratio. This is mainly because when the recycled coarse aggregate had a replacement rate of 100%, and the fiber content was the same, with the increase in its length–diameter ratio, the fiber length increased, and the fluidity of the mixture gradually decreased. At the same time, the “reservoir” effect of the recycled coarse aggregate further reduced the fluidity of the mixture, resulting in the difficult removal of bubbles coated in the mixture during the pouring process. Thus, a weak point was formed inside the specimen, which accelerated the destruction of the concrete and led to the reduction in compressive strength.

When the fiber volume fraction was greater than 0.10%, the saturated 3D mesh structure was gradually formed in the RAC due to the “bridging effect” of fibers, and the number of fibers decreased with the increase in the length–diameter ratio. Therefore, the 3D mesh structure tended to saturate first and then become unsaturated, and there was a critical saturation point. The above result suggests that the compressive strength of RAC increases first and then decreases.

#### 3.1.2. Splitting Tensile Strength

(1) Effect of fiber volume fraction on tensile strength

Figure 7 shows the relationship between the BFRAC splitting tensile strength and the BF volume fraction. As depicted in this figure, the influence law of the BF volume fraction on the splitting tensile strength of the BFRAC was consistent with that of ordinary concrete, and the change law of the BFRAC compressive strength with the BF volume fraction was consistent. The maximum increase rate of the RAC splitting tensile strength of different BF was clearly larger than that of ordinary concrete. The maximum increase rates of the splitting tensile strength of the BFRAC with 50% and 100% replacement rate were 32.84% and 31.61%, respectively. Gao’s [19] study showed that when the BF volume fraction was 0.15%, the increase rate of the splitting tensile strength could reach 44.8. The result of the comparison and analysis of the improvement rate of the BF on RAC compressive strength indicated that the improvement rate of the BF on RAC tensile strength was higher than that of compressive strength as a whole. This is mainly because under the action of a tensile load, the main force skeleton of concrete is hardened cement mortar, and the BF’s strengthening effect on concrete is essentially the strengthening of cement mortar. Therefore, tensile strength is better improved than compressive strength.

(2) Effect of fiber length–diameter ratio on tensile strength

Figure 8 shows the relationship between the BFRAC splitting tensile strength and the BF length–diameter ratio. As depicted in this figure, the tensile strength of the BFRC presented three trends with the increase in the fiber length–diameter ratio. In the first trend, when the BF volume fraction was 0.05%, 0.10% and 0.15%, the tensile strength of the BFRC increased first and then decreased. In the second trend, when the BF volume fraction was 0.20% and 0.25%, the tensile strength of the BFRC fluctuated. In the third trend, when the BF volume fraction was 0.30% and 0.35, the tensile strength of the BFRC decreased first and then increased. The effect of BF on the RAC splitting tensile strength is consistent with that of ordinary concrete, whereas the differences are as follows: The splitting tensile strength of BFRAC with 100% replacement ratio showed only two trends with the increase in the length–diameter ratio. Compared with ordinary concrete, these data lack the trend of first decreasing and then increasing, mainly because the number of micro-cracks in RAC is significantly higher than that in ordinary concrete. Therefore, there was a critical amount of fiber to constrain it. At the same fiber volume fraction, the number of fibers decreased with the increase in the length–diameter ratio. Thus, the constraint effect of fiber on RAC was not obvious. Nevertheless, due to the uneven strengthening effect of fiber, the stress concentration of micro-cracks in RAC would be caused, and the failure would be formed.

The result of the comprehensive analysis indicated that both the BF volume fraction and the length–diameter ratio affected the compressive strength and tensile strength of the BFRAC in unison, and the BF volume fraction had a more significant effect on the mechanical properties of concrete. Influenced by the dispersion performance of BF in concrete, when the volume fraction of BF did not exceed the optimal volume fraction, the 3D mesh structure formed in concrete was gradually saturated with the increase in the BF volume fraction, such that its compressive strength increased. When the volume fraction of the BF exceeded the optimal volume fraction, the probability of the BF clumping in concrete increased, thus affecting the compressive strength of concrete. As the BF continued to increase, the probability of fiber clumping also continued to increase. Subsequently, a large defect was formed inside the concrete, thus greatly reducing the strength of concrete.

When the BF volume fraction was small, the mechanical properties of concrete were mainly affected by the length–diameter ratio. The larger the length–diameter ratio, the longer the fiber length, and the more easily the fiber would exist in a curve shape in concrete. The bond property between fiber and cement mortar was optimized. With the further increase in the length–diameter ratio, the cross phenomenon easily appeared between the fibers, thus having an adverse effect on the concrete. However, when the volume fraction of the BF was large, since a certain length of fiber-base bond can ensure the full play of BF’s mechanical properties, as the fiber length–diameter ratio got larger, its length was also extended; the phenomenon of fiber crossing was also more likely to occur, thus having adverse effects on concrete. At the same volume fraction, when the length–diameter ratio was larger, the number of fibers was smaller. To achieve a certain number of 3D mesh structures, more fibers would be needed, thus leading to the increase in the optimal volume fraction.

#### 3.1.3. Tension–Compression Ratio

The tensile–compression ratio α_1_ can reflect the toughness and crack resistance of concrete materials. Figure 9 shows the scatter plot of the BFRAC’s tension and compression ratio under different replacement rates of recycled coarse aggregate. As depicted in Figure 9, the tension–compression ratio of the BFRC was smaller than that of the BFRAC, indicating that the effect of the BF on the RAC was better than that of the BF on concrete. This is because the recycled coarse aggregate has more pores and a rough surface which can increase the contact area with cement mortar and then increase the interface bond strength. In addition, in the process of RAC mixing, the mortar attached to the surface of the recycled coarse aggregate will quickly absorb water and combine with free water to form a film, and the absorbed water will be sealed inside. Thus, the “reservoir effect” is exerted [20]. 

During the curing phase, the sealing water is absorbed by the unhydrated cement particles, and the hydration reaction occurs, thus increasing the density of RAC. However, BF is distributed in a three-dimensional disorderly direction after RAC stirring. A mechanical meshing force and an adsorption adhesion force will be formed between the interface of fiber and the cement matrix [21], which further strengthens the interface transition zone between mortar and new mortar on the surface of the recycled coarse aggregate due to the bridging effect of the BF. Therefore, the tensile-compression ratio of the BFRAC is greater than that of the BFRC, and the toughness of the BFRAC is better than that of the BFRC. The analysis of test data indicated that the maximum improvement rates of the BFRC, the BFRAC-50% and the BFRAC-100% in the control group without BF were 6.49%, 14.18% and 14.95, respectively. It is worth noting that the BF characteristic parameters corresponding to the maximum increase rate of the tension–compression ratio are not consistent with the optimal volume fraction and the optimal aspect ratio. It can be seen from Figure 9 that when the BF is within the optimal volume fraction range, the tensile–compression ratio of the NC and the RAC mixed with the BF is smaller than that of plain concrete without the BF. Therefore, there are some shortcomings in the evaluation of BFRAC toughness using the tensile–compression ratio.

### 3.2. Characteristic Analysis of the Complete Stress–Strain Curve under Uniaxial Compression

#### 3.2.1. Failure Mode

The failure process of the RAC specimens is similar to that of the NC specimens. Under uniaxial compression, the crack development rate of the RAC and the NC specimens was fast. After the peak stress, the concrete specimen suddenly cracked into two independent blocks, showing brittle failure. However, the crack growth rate of the BFRC and BFRAC specimens was slow during loading. At the late loading stage, only a small quantity of concrete debris on the surface of the specimen flaked off. The specimen was still in its whole state at the last moment of loading, showing obvious ductility. From the perspective of the crack development process, the failure mode of the NC and the RAC was oblique shear failure, which developed from the corners of both ends to the middle of the specimen. The angle between the failure surface and the load vertical line was nearly 60~70°, while the angle between the failure surface and the load vertical line was more than 75° for the BFRAC and the BFRC. After the specimen was destroyed, the surface micro-cracks of the concrete with BF were significantly more than those without BF, and the maximum crack width was significantly smaller than that of the concrete without BF. The failure sections of the BFRC and the BFRAC specimens indicated that the BF was mainly pulled out from the concrete, and the “crackling” sound of BF was heard during the failure process.

The distribution of cracks on the surface of the specimen indicated that with the increase in the fiber volume fraction, the width of the main crack on the surface of the specimen tended to decrease, while the number of micro-cracks gradually increased. This is because BF can reduce the probability of micro-cracks forming and spreading in the base when the specimen is subjected to uniaxial longitudinal pressure. When the micro-cracks in concrete began to extend and expand under the vertical load, overlapped BF at both ends of the cracks acted as a bridge, so as to inhibit and constrain crack growth. With the increase in the fiber volume fraction, the three-dimensional distribution range of fiber in concrete was wider, and the inhibition effect of fiber on the development of micro-cracks in concrete was stronger. With the increase in the macro-crack width, the microfiber was constantly pulled out and broken, which effectively delayed and prevented the crack expansion. Therefore, the specimen exhibited an increase in fine cracks, while the width of the main crack decreased.

With the increase in the length–diameter ratio, the formation velocity of the main cracks and the total number of cracks on the surface of the BFRC specimens first decreased and then increased. The main reason is that when the BF’s length–diameter ratio is less than the optimal length–diameter ratio with the same fiber content, the fiber anchorage length at both ends of the micro-crack increases with the increase in the length–diameter ratio, and the inhibition effect on micro-crack growth is enhanced. When BF length–diameter ratio is greater than the optimal length–diameter ratio, the fluidity of the mixture decreases gradually with the increase in the length–diameter ratio. Therefore, it is difficult to remove the bubbles coated in the mixture during the casting process, and many weak spots are formed in the specimen. Under the action of a compression load, the weak point will become the inducing factor for crack development, so the total number of cracks increases. However, due to the increase in initial micro-cracks, the probability of forming a penetrating main crack increases, thus resulting in the rapid formation of the main crack.

#### 3.2.2. Feature Points of the Complete Curve

The peak point of the stress–strain curve of concrete most directly indicates the bearing capacity and deformation capacity of concrete before failure. Different feature points on the stress–strain curve indicate the distribution, expansion and fracture of cracks in concrete [22,23].

(1) Peak stress

Figure 10 shows the curve of peak stress versus the volume fraction and the length–diameter ratio of BFRAC. As depicted in Figure 10a,c,e, the peak stress of BFRAC increased first and then decreased with the increase in the fiber volume fraction; there was an optimal volume fraction (0.20–0.25%) of peak stress of BFRAC with different length–diameter ratios. When the volume fraction was the optimal volume fraction, the peak stress of the BFRAC-50% and the BFRAC-100% increased by nearly 17.42~23.15% and 16.09~26.94% when compared with the RAC, respectively, and the peak stress of the BFRC increased by nearly 2.17~6.32% when compared with that of ordinary concrete under different fiber length–diameter ratios. The above results indicated that the effect of the BF on the peak stress of the RAC under uniaxial compression was significantly better than that of the NC. The main reason for this phenomenon is that the BF strengthens the interface transition zone between the surface adhesion mortar and the internal micro-cracks of the recycled coarse aggregate and the new cement mortar. In ordinary concrete, however, the BF only strengthens cement mortar.

As depicted in Figure 10b,d,f, when the fiber volume fraction was 0.05%, 0.10% and 0.15%, the effect of the BF length–diameter ratio on the peak stress of concrete can be neglected, since the total amount of fiber in concrete is small and the saturated 3D mesh structure has not yet been formed. At the same volume fraction, the fiber length increased, the number of fibers decreased, and the three-dimensional mesh structure became more unsaturated with the increase in the length–diameter ratio. Therefore, the strengthening effect on concrete was not significant. When the fiber volume fraction was 0.20~0.25%, the amount of fiber was sufficient, and the 3D mesh structure formed in the concrete which reached the saturation state. With the increase in the fiber length–diameter ratio, it increased first and then decreased, and the optimal length–diameter ratio was 1200~1400. Original concrete had a length–diameter ratio of 1200, while the RAC had a length–diameter ratio of 1400. This is because the recycled coarse aggregate in the RCA has more porosity and micro-cracks, and the interface transition zone between the RCA and the new cement mortar is thicker, so the bridging effect of longer BF can fully take place.

(2) Peak strain

Peak strain is the strain corresponding to peak stress. Figure 11 shows the curve of the BFRC peak strain versus the volume fraction and the length–diameter ratio. As depicted in Figure 11a,c,e, the peak strain of concrete increased first and then decreased with the increase in the fiber volume fraction under different length–diameter ratios. There was an optimal volume fraction of peak stress for different length–diameter ratios. When the fiber volume fraction reached the optimal volume fraction, the peak strain of the BFRAC-50% and the BFRAC-100% increased by nearly 25.65–29.03% and 26.28–29.15% when compared with the RAC, respectively, while the peak strain of the BFRC increased by nearly 23.12–26.40% when compared with that of ordinary concrete. It can be seen that the incorporation of BF can significantly improve the deformation ability of concrete under axial compression. Notably, the increase rate of the BF optimal volume fraction to the peak strain of concrete was significantly larger than that of the BF to the peak stress. The reason for this phenomenon needs to be analyzed from the AB section of the complete stress–strain curve. In section AB, the surface of the BFRC specimen showed fine cracks. However, the strain of the BFRC was greater than that of the NC due to the co-loading of the BF and the concrete base. In addition, the main reason why BF increases RAC peak strain slightly more than ordinary concrete is that the elastic modulus of recycled coarse aggregate is lower than that of natural coarse aggregate, consistent with the research conclusion of Xiao Jianzhuang [24].

As depicted in Figure 11b,d,f, on the whole, the peak strain of BFRAC increased first and then decreased with the increase in the length–diameter ratio. The optimum length–diameter ratio was in the peak strain of concrete under different BF volume fractions. When the fiber volume fraction was the same, the BF length–diameter ratio had little influence on the peak strain of concrete. When the length–diameter ratio did not reach the optimal length–diameter ratio, the improvement rates of the BFRC, the BFRAC-50% and the BFRAC-100% were 0.94–6.23%, 0.73–6.24% and 1.72–6.19%, respectively. It is worth noting that the fiber length to diameter ratio on the increase rate of the peak strain of concrete is lower than the fiber volume fraction rate of the ascent of the peak strain of concrete; it shows that the number of fibers is the main factor influencing the peak strain of concrete. The fiber in the concrete in the form of a saturated three-dimensional grid structure is the essential reason for ascension of concrete ductility. However, when the fiber length–diameter ratio meets a certain anchorage length, the bond performance between the fiber and the concrete base is the premise to ensure that the fiber gives full play to its strengthening and toughening. By analyzing the microstructure of BFRC, Lin Jiafu [25] found that BF formed a dense three-dimensional spatial network structure inside the concrete, which greatly improved the mechanical stability of the micro-area of the concrete interface and inhibited the generation and development of micro-cracks in the concrete interface.

#### 3.2.3. Curve of Feature

Figure 12, Figure 13 and Figure 14 show the complete stress–strain curves of the BFRC, the BFRAC-50% and the BFRAC-100% under uniaxial compression with the change in BF volume fraction under different fiber length–diameter ratios, respectively. As depicted in Figure 12, Figure 13 and Figure 14, the fiber length–diameter ratio, the fiber volume fraction and the recycled coarse aggregate replacement ratio had little influence on the shape of the stress–strain curve of concrete but had clear influence on the slope of the curve.

Figure 15 shows the stress–strain curves of the BFRC specimens under uniaxial compression with different fiber volume fractions. The whole process of BFRC axial compression can be divided into four stages (Figure 16).

The OA stage was in the linear elastic stage, and there was no crack on the surface of the specimen. In this stage, the BF’s toughening and cracking resistance was not fully at work and was close to that of ordinary concrete.

The AB stage was the meso-crack propagation stage. The initial crack inside the specimen gradually expanded with the increase in load, and small cracks parallel to the loading direction appeared on the surface of the specimen. When the load exceeded 80% of the peak stress, relatively clear macroscopic cracks appeared on the surface of the specimen. When the load reached the peak stress, the macroscopic cracks on the surface of the specimen gradually expanded, gathered and crossed to form macroscopic main cracks. At this stage, the BF crack resistance effect was obvious, which can inhibit the development of micro-cracks and the generation of new cracks in concrete.

The BC stage was the macro-fracture stable extension section. After reaching peak stress, the stress began to decrease as the load continued to increase. This phenomenon of stress reduction with deformation increase, namely strain softening, is an important feature of common concrete and BFRC under uniaxial compression. Compared with ordinary concrete, the number and development rate of BFRC cracks increased.

The CD stage was the fracture instability failure stage. As the strain continued to increase, the stress decreased slowly, and the macroscopic main crack on the surface of the specimen gradually formed a penetrating crack; crack width increased with the increase in load. At this stage, the BF mainly played a bridging role by mostly pulling out.

In the OA stage, the stress and strain of the BFRAC and the RAC increased linearly, whereas the stress–strain curve of BFRAC was basically on the left of RAC, thus indicating that the stiffness of BFRAC was greater than that of RAC. At the AB stage, the stress increased steadily while the strain increased at a higher rate, and the slope of the curve gradually decreased; the stiffness of the specimen degraded. With the increase in axial deformation, the stress–strain curve of the BFRAC was still on the left of the RAC. At this stage, the effect of the BF on RAC crack resistance began to appear. At the BC stage, after the RAC reached the peak load, the specimens failed rapidly and the load dropped sharply, and the RAC showed the characteristics of brittle failure. At this stage, the rate of decline of the BFRAC curve was relatively slow, and the stress–strain curve was located on the upper side of the RAC. The main reason is that, at this stage, fiber and concrete worked together in coordination. The BF played a role in the form of bridge stress, across the crack interface transfer stress, thus effectively avoiding the phenomenon of “one crack, two stages”. In the CD stage, the stress–strain curve of the BFRAC decreased gradually and became flat. At this stage, the BF was pulled out and broken into large quantities, and the specimen was seriously damaged. Meanwhile, RAC and NC showed no flat segment, which showed a split and two segments in macro.

### 3.3. Establishment of a Complete Stress–Strain Curve Model

#### 3.3.1. Classical Model

The stress–strain curve of concrete under uniaxial stress lays an experimental data basis for structural design and nonlinear analysis [26]. Scholars worldwide have proposed a series of uniaxial compression stress–strain curve models of single-doped fiber concrete and hybrid fiber concrete (Table 6). As depicted in Table 6, the constitutive models of fiber reinforced concrete at home and abroad are mainly obtained by revising the constitutive models of ordinary concrete. The above models cannot be applied to BFRC due to the different types of fibers, and the effects of fiber volume content and fiber length–diameter ratio are not considered.

The complete stress–strain curve of BFRC was predicted using the three models of Guo Zhenhai [23] (Model 1), Mehran [31] (Model 2) and the European standard [32] (Model 3), and compared with the test curve, as shown in Figure 17. The comparison between the three models and the test data indicated that the three models were in good agreement with the test results in the ascending stage. However, in the descending stage, the prediction effects of different models were quite different. The Guozhenhai model was in good agreement with the test curve, while the European standard model had a large error.

#### 3.3.2. Establishment of Models

At present, the stress–strain curve equation of concrete is generally expressed in a dimensionless way, that is, $x=\varepsilon /\varepsilon_{0},y=\sigma /\sigma_{0}$ ($\varepsilon_{0}$ is the strain corresponding to the peak stress, and $\sigma_{0}$ is the peak stress of the curve). The existing concrete stress–strain curves all need to meet the following four boundary conditions:(1)The curve passes through the origin and point of coordinates.(2)At (1,1) the slope is 0.(3)x→∞, y→0.(4)0 ≤ y ≤ 1.

In this paper, the sectional equation proposed by Guo Zhenhai was adopted, and the equation of the ascending section of the BFRAC compression stress-strain curve is
(1)$${y=ax+(3-2a)x}^{2}{+(a-2)x}^{3}\;\;0\leq x\leq 1.$$

The equation of the descending section is expressed as
(2)$$y=\frac{x}{{b(x-1)}^{2}+x}\;\;x>1.$$

According to the analysis in Section 3.1, the BF length–diameter ratio has little influence on the relevant mechanical properties of concrete, such that the optimal length–diameter ratio of 1200 is taken as an example. The Nonlinear Cur Fit function in Origin software was used to fit the stress–strain test data of BFRC, BFRAC-50% and BFRAC-100% axial compression. The fitting results are shown in Figure 18. As depicted in Figure 18, the BF had a small impact on the ascending section of the RAC, but a large impact on the descending section.

Parameter a represents the ratio of the initial elastic modulus of concrete axial compression to the peak secant modulus [27]. Figure 19 shows the curve of a value of the BFRAC. As depicted in Figure 19, the *a* value increased first and then decreased with the increase in the fiber volume fraction, and the overall change range was small. The *a* values of the BFRC, the BFRAC-50% and the BFRAC-100% were 2.08–2.27, 2.04–2.18 and 1.96–2.12, respectively. Therefore, the average *a* values of the BFRC, the BFRAC-50% and the BFRAC-100% were 2.19, 2.12 and 2.06. Figure 20 shows the *b* value change curve of the BFRAC. As depicted in Figure 20, the *b* value decreased first and then increased with the increase in the fiber volume fraction. The distribution range of the *b* value was 0.78~1.53, and the variation range was larger than that of the *a* value on the whole, thus proving that the effect of the BF on the descending section of the RAC was greater than that on the ascending section.

In order to better describe the effect of the BF on the RAC stress–strain, a fiber strengthening factor *F* [7,8,33,34] considering the BF volume fraction and the fiber geometric characteristics was introduced in this paper, which is written as follows:(3)$${F=\beta V}_{f}\frac{l_{f}}{d_{f}}$$
where *β* is the fiber bond coefficient; *V*_f_ is fiber volume fraction; *l*_f_ is fiber length; *d*_f_ is the fiber diameter.

The bond coefficient is related to the geometric characteristics of the fiber: since BF is a flexible fiber it mostly exists in the form of curve in concrete. Compared with the round straight fiber, the bond performance is better. Therefore, the BF bond coefficient was set at 0.75 [35]. The relationship between parameters *b* and *F* was obtained through data regression analysis, and the results are listed in Table 7.

By substituting Equations (4)–(6) into Equation (2), the constitutive model of the descending section of the stress–strain curve of BFRAC under uniaxial compression load can be obtained.

### 3.4. BFRAC Toughness Evaluation

Axial compression toughness reflects the ability of concrete to absorb energy when it continues to bear the load after cracking. Fracture energy is an important toughness index for concrete [7]. In order to quantitatively evaluate the influence of the BF volume fraction and the aspect ratio on the toughness of concrete under compression load, the BFRAC uniaxial compression load–deformation curve was obtained through tests in this paper, and the region (*T*) surrounded by the curve and coordinate axis is defined as the fracture energy of uniaxial compression failure [36,37], which characterizes the compression toughness of concrete. *T* can be obtained from Equation (7) [38].
(7)$$T=\int_{0}^{1}{ax+(3-2a)x}^{2}+\left(a-2\right)x^{3}\mathrm{dx}+\int_{1}^{6}\frac{x}{{b(x-1)}^{2}+x}\mathrm{dx}$$

The *T* values of each group can be obtained from Equation (7) as shown in Table 8.

It can be seen that the toughness of the BFRAC under uniaxial compression increased first and then decreased with the increase in the fiber strengthening factor. When the fiber strengthening factor was 2.4, the fiber volume fraction 0.20% and the aspect ratio 1200, the toughness of the BFRC, the BFRAC-50% and the BFRAC-100% increased by 11.32%, 21.96 and 26.59, respectively, when compared with the control group.

The analysis of the tension–compression ratio and fracture energy indicated that the tension–compression ratio increased first and then decreased with the change in BF volume fraction; this was consistent with the effect of the BF on the basic mechanical properties of RAC, whereas the corresponding optimal fiber volume fraction was different. According to the fracture energy index, the maximum increase rate of BFRAC toughness was higher than that of the control group, which was higher than that of the tensile–compression ratio. The result of the comprehensive analysis of the influence law and the increase effect of the BF on the compressive properties of the RAC reveals that fracture energy is more suitable for evaluating the compressive toughness of BFRAC.

## 4. Conclusions

The paper focuses on the study of the basic mechanical properties and the uniaxial compression constitutive relationship of basalt fiber recycled concrete. It discusses the influence of the volume fraction and the length–diameter ratio of basalt fiber on mechanical properties, including the axial compression strength, peak strain and compression toughness. It also establishes a theoretical full curve model combined with current research and the characteristics of basalt fiber recycled concrete. The compression toughness of BFRAC is quantitatively evaluated by the tension–compression ratio and the uniaxial compression failure fracture energy. In accordance with the calculation and analysis results, the following conclusions were obtained:

(1) Basalt fiber (BF) can effectively improve the compressive strength and splitting tensile strength of recycled aggregate concrete (RAC). The compressive strength of basalt fiber-reinforced recycled aggregate concrete (BFRAC) with 50% and 100% replacement ratios was 18.31% and 24.0%, higher than that of RAC without fiber, respectively. The maximum increase rates of the splitting tensile strength were 32.84% and 31.61%, respectively.

(2) The peak stress and peak strain of the RAC increased first and then decreased with the increase in the BF volume fraction. The effect of the BF length–diameter ratio on RAC peak stress and peak strain was small. The peak stress of the BFRAC with 50% and 100% replacement rates had maximum increase rates of 23.15% and 26.94%, respectively, and the peak strain of this concrete had maximum increase rates of 29.03% and 29.15%, respectively.

(3) Based on the stress–strain curve analysis of the BFRAC uniaxial compression, a fiber strengthening factor considering the influence of the fiber length–diameter ratio and the volume fraction was introduced to establish the stress–strain curve expression of the BFRAC uniaxial compression. The correlation coefficient was more than 0.85, which provides a theoretical basis for the engineering application of BFRAC.

(4) Compared with the tensile–compression ratio, the compressive failure fracture is more suitable for evaluating the toughness of BFRAC under compression. Compared with the control group, the toughness of the BFRAC with 50% and 100% replacement rates increased by 21.96% and 26.59%, respectively. BF has a good effect on improving the toughness of recycled concrete, which is helpful to realize the large-scale application of recycled concrete.

## Figures and Tables

**Figure 1 materials-16-01849-f001:**
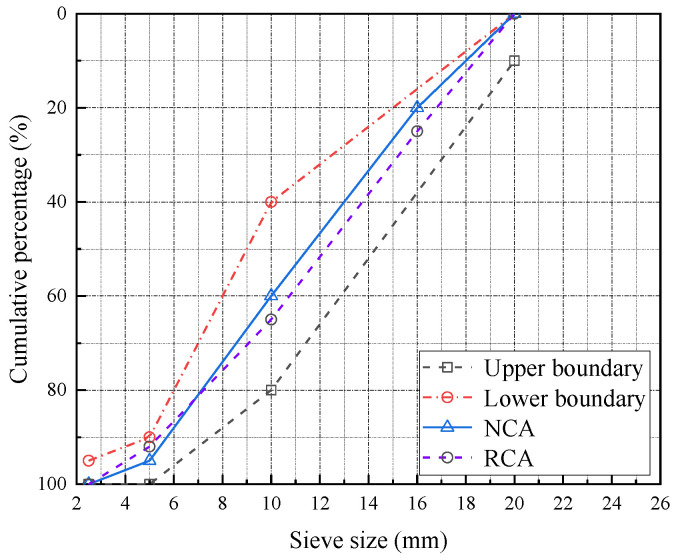
Gradation of coarse aggregate.

**Figure 2 materials-16-01849-f002:**
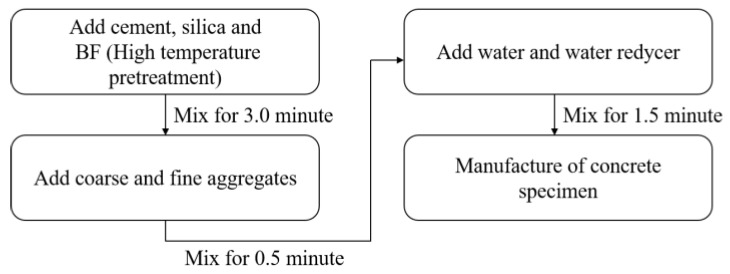
Mixing procedures of BFRAC.

**Figure 3 materials-16-01849-f003:**
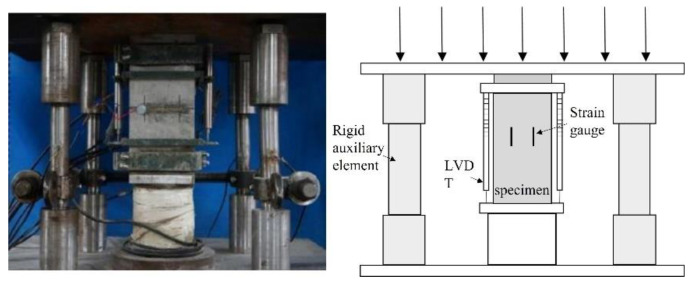
Uniaxial compression test device.

**Figure 4 materials-16-01849-f004:**
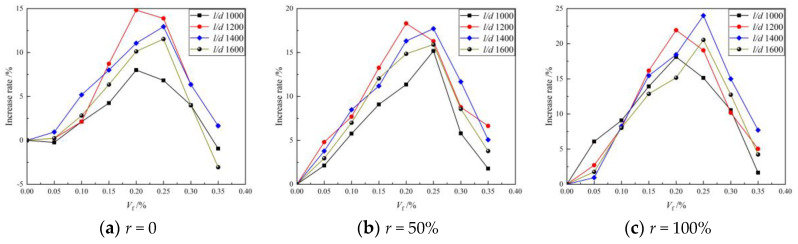
Relation between BFRAC cube compressive strength and BF volume fraction.

**Figure 5 materials-16-01849-f005:**
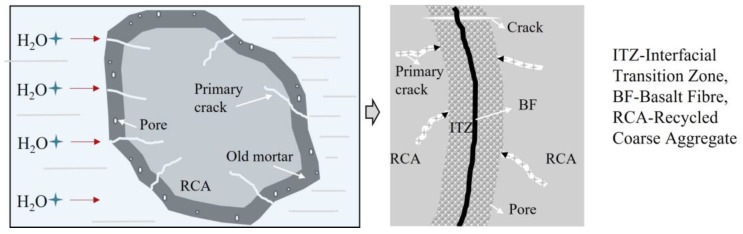
Schematic Diagram of RCA’s “Reservoir Effect”.

**Figure 6 materials-16-01849-f006:**
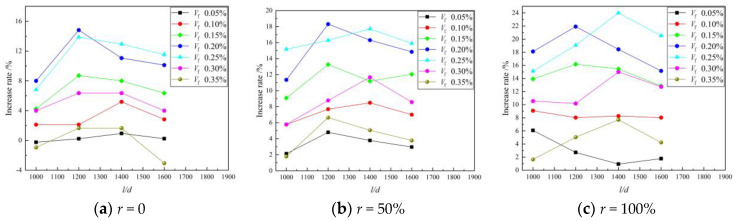
Relation between BFRAC cube compressive strength and BF length–diameter ratio.

**Figure 7 materials-16-01849-f007:**
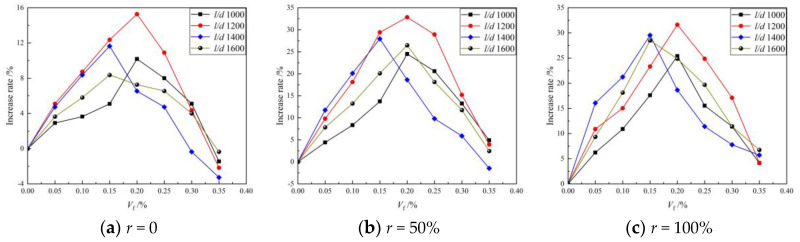
Relationship between BFRC splitting tensile strength and BF volume fraction.

**Figure 8 materials-16-01849-f008:**
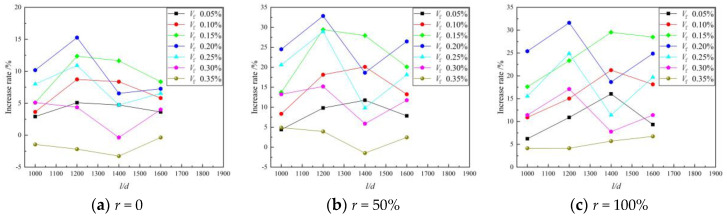
Relationship between BFRAC splitting tensile strength and BF length–diameter ratio.

**Figure 9 materials-16-01849-f009:**
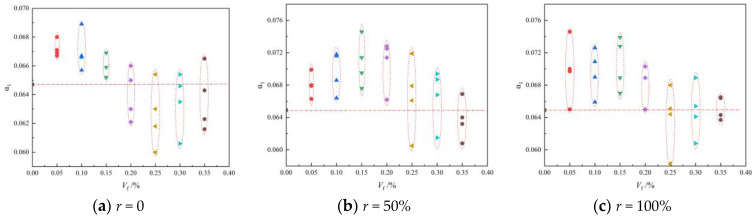
Scatter diagram of BFRAC tension compression ratio under different recycled coarse aggregate replacement rates.

**Figure 10 materials-16-01849-f010:**
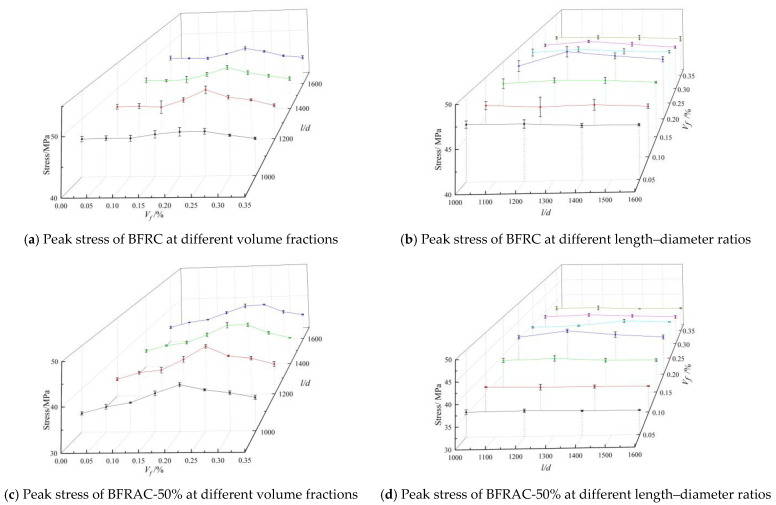
Peak stress of BFRAC under uniaxial compression.

**Figure 11 materials-16-01849-f011:**
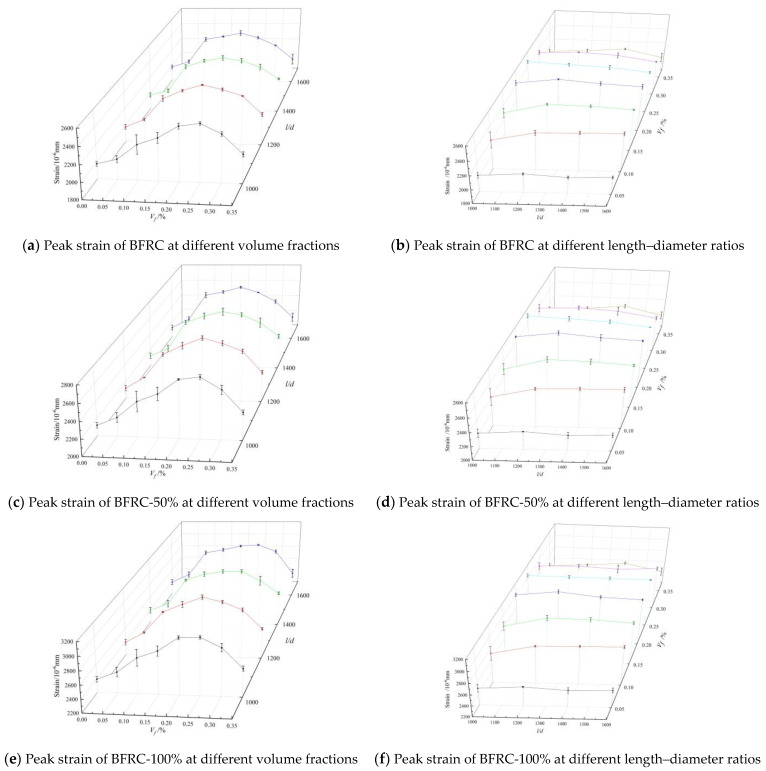
Peak strain of BFRAC under uniaxial compression.

**Figure 12 materials-16-01849-f012:**
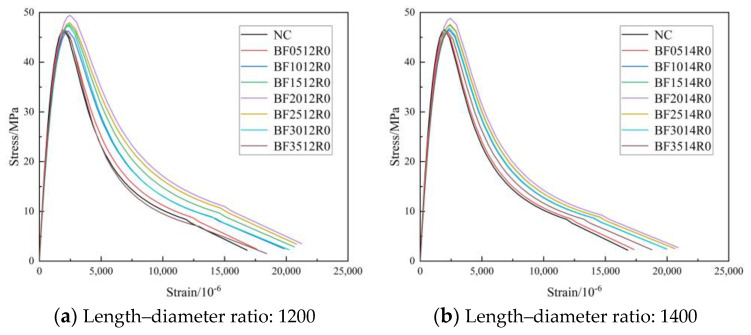
Complete stress-strain curve of BFRC under uniaxial compression.

**Figure 13 materials-16-01849-f013:**
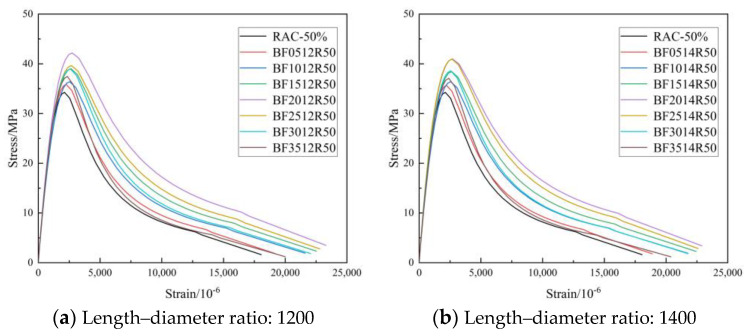
Complete stress–strain curve of BFRAC-50% under uniaxial compression.

**Figure 14 materials-16-01849-f014:**
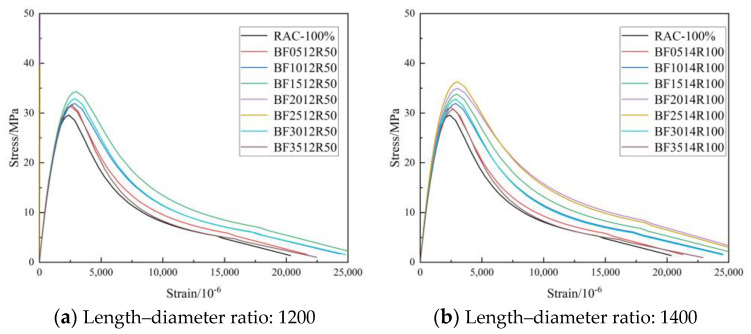
Complete stress–strain curve of BFRAC-100% under uniaxial compression.

**Figure 15 materials-16-01849-f015:**
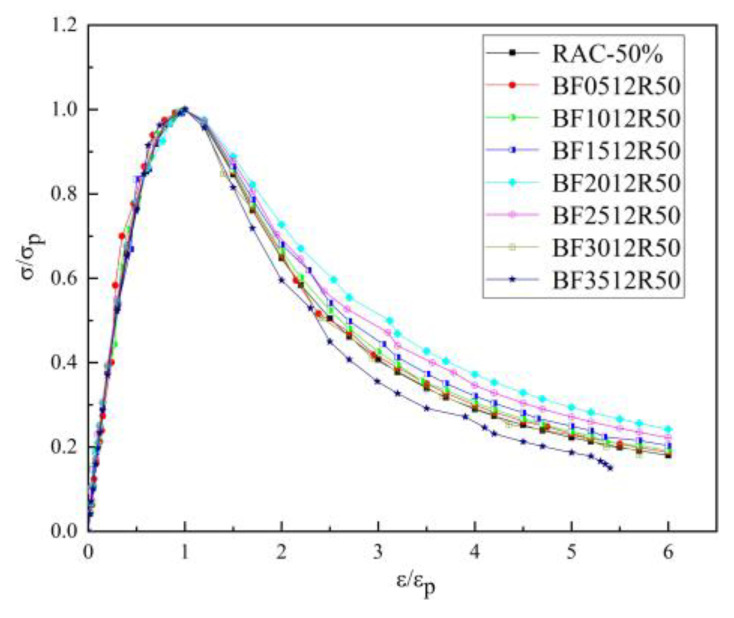
Full stress–strain curve of BFRC under compression.

**Figure 16 materials-16-01849-f016:**
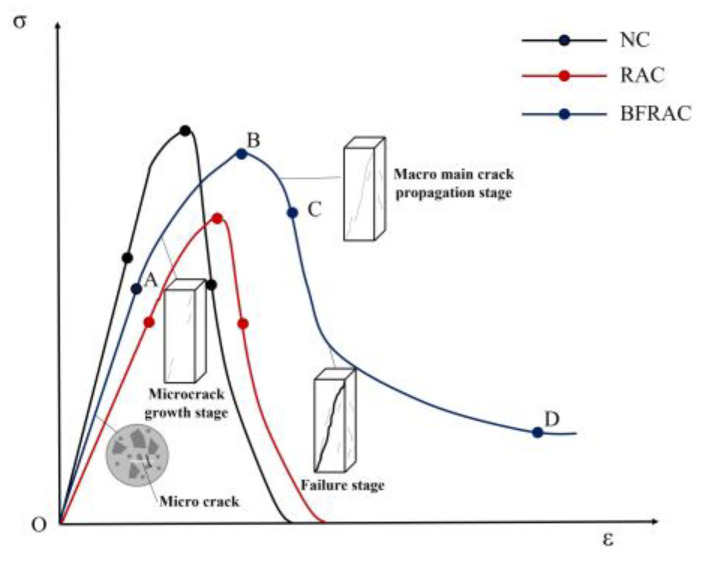
Whole process of BFRC under compression.

**Figure 17 materials-16-01849-f017:**
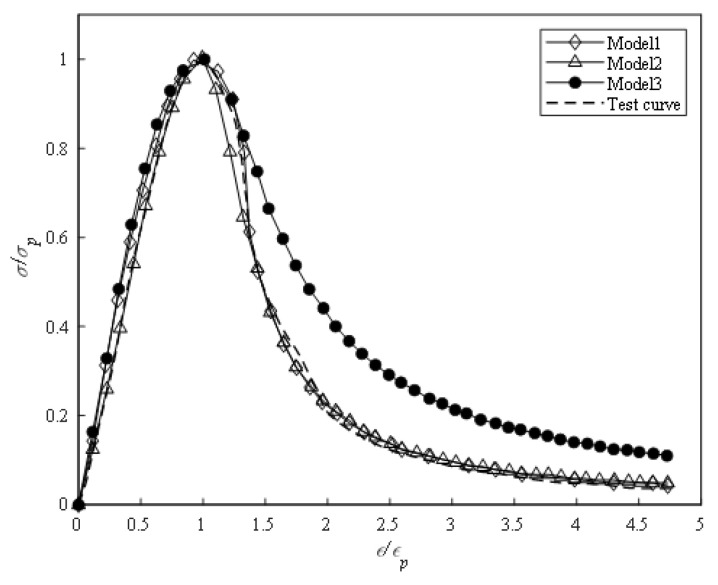
Prediction and test results.

**Figure 18 materials-16-01849-f018:**
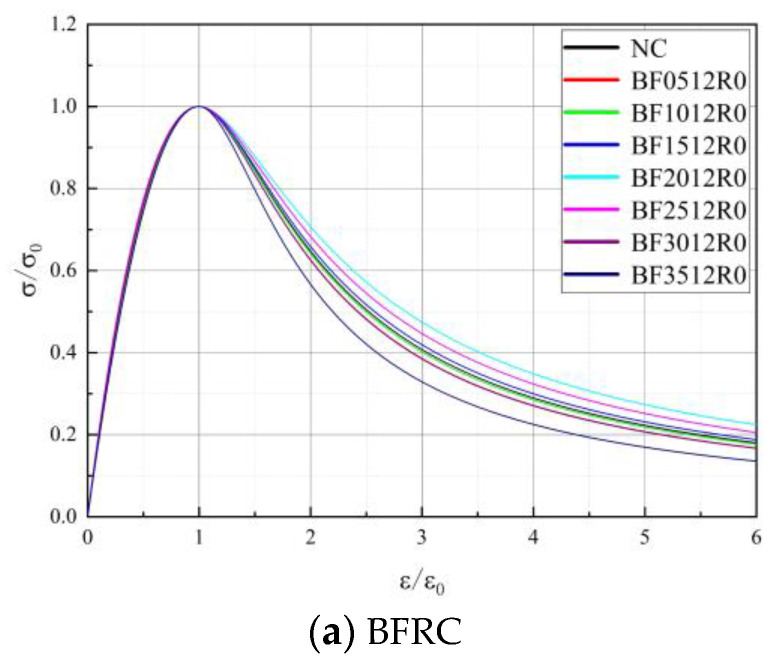
Axial compressive stress–strain curve fitting of BFRAC.

**Figure 19 materials-16-01849-f019:**
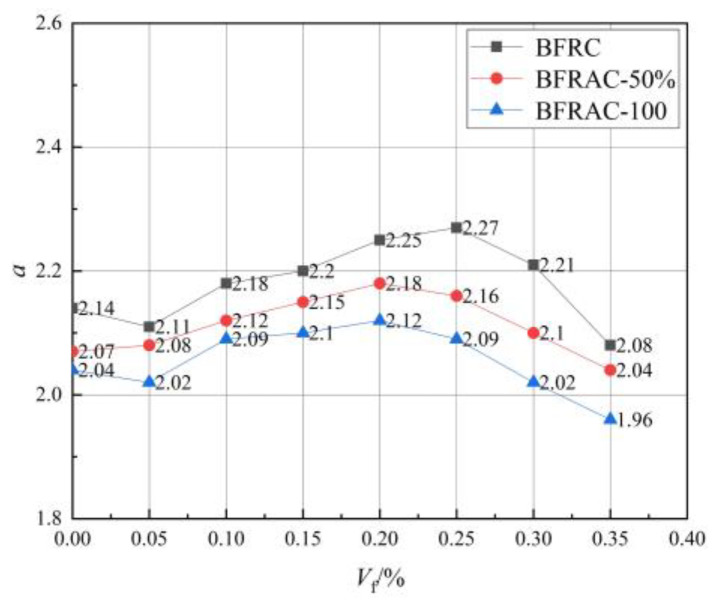
*a*-value change curve of BFRAC.

**Figure 20 materials-16-01849-f020:**
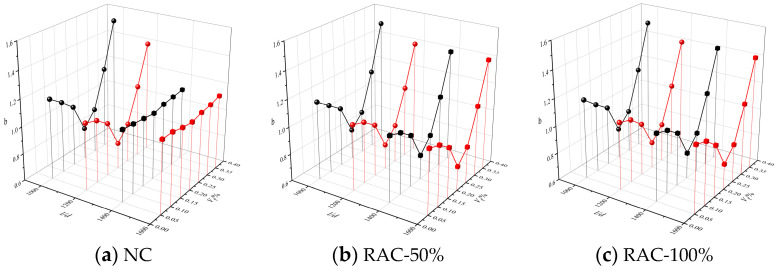
*b*-value change curve of BFRAC.

**Table 1 materials-16-01849-t001:** Main chemical composition and physical properties of cement and SF.

Index	Cement	SF
SiO_2_/%	20.88	94.26
Al_2_O_3_/%	2.86	0.78
Fe_2_O_3_/%	4.66	0.66
CaCl_2_/%	—	—
MgCl_2_/%	—	—
Na_2_O/%	0.48	0.58
MgO/%	1.66	0.64
K_2_O/%	0.26	2.65
CaO/%	69.04	0.43
MnO/%	0.16	—
Specific gravity/(kg·m^−3^)	3120	2310
Specific surface area/(m^2^·kg^−1^)	334	18,954
Loss on ignition/%	1.5	2.28

**Table 2 materials-16-01849-t002:** Physical properties of aggregates.

Parameters	NCA	RCA	S
Apparent density/(kg/m^3^)	2660	2430	2720
Bulk density/(kg/m^3^)	1430	1170	1450
Water absorption/%	0.63	2.50	1.20
Water absorption of 24 h/%	1.19	5.24	2.80
Crush index	11.70	14.48	—
Needle and flake particle content	2.24	1.35	—
Clay lump content	0.6	0.2	0.12
Fineness modulus	—	—	2.6

**Table 3 materials-16-01849-t003:** Physical properties of basalt fiber.

Length/mm	Density/(kg/m^3^)	Tensile Strength/MPa	Elasticity Modulus/GPa	Elongation at Break/%
15~24	2650	3800~4840	79.3~93.1	3.1

**Table 4 materials-16-01849-t004:** Mix proportion of the concrete.

**Mix**	**BF** **(%)**	**CA** **(kg/m^3^)**	**S** **(kg/m^3^)**	**Cement** **(kg/m^3^)**	**SF** **(kg/m^3^)**	**Water** **(kg/m^3^)**	**SP** **(kg/m^3^)**
NC	0	1211	682	364	27	195	1.5

**Table 5 materials-16-01849-t005:** Specimen number.

Specimen Number	*V*_f_/%	*l*/*d*	*r*/%
BF0510R0	0.05	1000	0
BF0510R50	0.10	1000	50
BF0510R100	0.15	1000	100

Note: Take BF0510R50 as an example, where BF represents basalt fiber, 05 represents volume fraction of 0.05%, 10 represents length–diameter ratio of 1000, R represents RAC and 50 represents replacement rate of recycled coarse aggregate of 50%.

**Table 6 materials-16-01849-t006:** Classical model of stress–strain curve under uniaxial compression.

Constitutive Model	Type of Fiber	Parameters of Fiber	References
Ascending section: ${y=ax+(3-2a)x}^{2}{\;+\;(a-2)x}^{3}$ Descending section: $y=\frac{x}{b{\left(x\;-\;1\right)}^{2}\;+\;x}$	Steel–polypropylene fiber	${F=\beta V}_{f}\frac{l_{f}}{d_{f}}$	Zhang Yuanyuan [27]
Ascending section: ${y=ax+(3-2a)x}^{2}{\;+\;(a-2)x}^{3}$ Descending section: $y=\frac{k_{1}\beta x}{k_{1}{\beta \;-\;1\;+\;x}^{k_{2}\beta }}$	Steel–carbon fiber	*V* _f_	Wei Hui [28]
Ascending section: $y=\frac{a_{0}{x\;-\;x}^{2}}{{1\;+\;(a}_{0}\;-\;2)x}$ Descending section: $y=\frac{x}{{b(x\;-\;1)}^{2}\;+\;x}$	Steel fiber	-	Bing Liu [29]
Ascending section: ${y=ax+(5-4a)x}^{4}{\;+\;(3a-4)x}^{5}$ Descending section: $y=\frac{x}{{b(x\;-\;1)}^{2}\;+\;x}$	Steel–basalt fiber	-	Mehran Khan [30]

**Table 7 materials-16-01849-t007:** The fitting equation between *b* and *F*.

Concrete	Fitted Equation	R^2^	Equation No.
BFRC	$b=1.0897\;+\;0.1766F\;-\;{0.3554F}^{2}+{0.1061F}^{3}$	0.8859	(4)
BFRAC-50%	$b=1.1238\;+\;0.0871F\;-\;{0.3064F}^{2}+{0.0984F}^{3}$	0.8929	(5)
BFRAC-100%	$b=1.1344\;+\;0.0854F\;-\;{0.3093F}^{2}+{0.0995\sigma }^{3}$	0.9006	(6)

**Table 8 materials-16-01849-t008:** *T* value.

Specimen Number	*T*	Specimen Number	*T*	Specimen Number	*T*
NC	2.828	RAC-50%	2.587	RAC-100%	2.478
BFRC05-1200	2.826	BFRC05-1200	2.832	BFRC05-1200	2.818
BFRC10-1200	2.858	BFRC10-1200	2.862	BFRC10-1200	2.850
BFRC15-1200	2.924	BFRC15-1200	2.930	BFRC15-1200	2.926
BFRC20-1200	3.148	BFRC20-1200	3.155	BFRC20-1200	3.137
BFRC25-1200	3.033	BFRC25-1200	3.035	BFRC25-1200	3.029
BFRC30-1200	2.800	BFRC30-1200	2.799	BFRC30-1200	2.784
BFRC35-1200	2.588	BFRC35-1200	2.585	BFRC35-1200	2.572

## Data Availability

Not applicable.

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
