# Peer review of "Effect of Basalt Fiber on Uniaxial Compression-Related Constitutive Relation and Compressive Toughness of Recycled Aggregate Concrete"

_materials, 2023, doi:10.3390/ma16051849_

Round 1

Reviewer 1 Report

The Abstract is too long. It should be reduced to almost half.

The length-diameter ratio is defined in Table 4 as l/d, while it is defined in the legends of Figure 5 and 8 as l/f.

How did the authors calculate the toughness in the basalt fiber reinforced concrete (using equation 6), and which standard (i.e. ASTM, European, etc.) did the follow to do that?

What test did the author use the calculate the toughness? Third point loading flexural test is to be used to calculate the toughness.

Reviewer 2 Report

It looks well organized and I believe it is good enough to be published. A few minor comments are below: 

1) Fig. 22 and Table 8: R^2 is low. Is there other way to increase the R^2? In addition, BFRC fitting curve is different from the other two because of the initial data point. Not sure if the authors explained of this. The T values for BFRC does vary much in the first a few F points while others are varying significantly.

2) Please add one paragraph summary in the beginning of conclusion before listing the bullets.

3) Where does Eq 6 come from? How did the authors develop that equation for T?

Reviewer 3 Report

First, it does not fit the structure of the article at this point.

Currently there is a lot (too much) data, authors need to reduce the amount of data by presenting only critical values. As an example: what are the differences between: Fig. 13-15: c and d parts. The authors describe/comment on Figures 13-15 in 2 sentences (128 curves comment in 2 sentences). Also, what's these curves practical significance of the scientific research? Why all these curves presented in this paper? The same tendency is observed in other results parts.

I do not see scientific value in these figures (Fig. 2 and Fig. 10). They need to be removed.

The part “Fracture energy”: first of all, your equation is not reliable (as shown by R2 values). Figure 22: Initial data (only one value is different for you, when the x-axis value is equal to 1; other values are the same) - what is the purpose of computing mathematical models where only one value varies? Why this negative result is given? (What is the scientific meaning/added value?)

The results are discussed, I miss a little any comparison with the other authors how used basalt fiber with recycled aggregate concrete.

Conclusion also needs to be rewritten. Include the following: new concepts and innovations demonstrated in this study, summary of findings, comparison with findings by other workers, and concluding remark (what’s new?).

Reviewer 4 Report

authors carried out an extensive research and an indepth analysis. However the following needs to be addressed

·         At the end of the conclusion please clearly state your research problem, research gap and objective

·         Include a Table for the chemical properties of silica fume and cement and explain

·         How was the mix proportion in Table 3 achieved? Which standard or procedure was followed? Please explain step by step

·         Why is only one proportion of silica fume used?

·         Only compressive strength test was performed? It would have been good if modulus of elasticity test was done

·         You result in section 3.1 needs to be compared with previous studies since there are related studies available

·         The conclusions needs to be shortened

Reviewer 5 Report

The study has both a scientific effect and an industrial contribution. In reference selection, the current study can be compared with the studies conducted in the last 5 years. Minor revisions can be made in terms of language and expression.

Round 2

Reviewer 1 Report

The comments were responded to satisfactorily. However, the response to comment 3 should be implemented in the manuscript.

Author Response

Thank you very much for your comments. 

Reviewer 3 Report

corrections are appropriate, I suggest to accept

Author Response

Thank you very much for your comments.